# Antimicrobial Mechanism of pBD2 against *Staphylococcus aureus*

**DOI:** 10.3390/molecules25153513

**Published:** 2020-07-31

**Authors:** Kun Zhang, Heng Zhang, Chunyu Gao, Ruibo Chen, Chunli Li

**Affiliations:** College of Animal Science and Veterinary Medicine, Henan Agricultural University, Zhengzhou 450002, China; henauzk@163.com (K.Z.); zh904252860@163.com (H.Z.); cygao89@126.com (C.G.); chen.rui.bo.happy@163.com (R.C.)

**Keywords:** antimicrobial peptides, antimicrobial mechanism, morphological changes, differentially expressed genes, electron microscopy, quantitative real-time PCR

## Abstract

Antimicrobial peptides (AMPs) show high antibacterial activity against pathogens, which makes them potential new therapeutics to prevent and cure diseases. Porcine beta defensin 2 (pBD2) is a newly discovered AMP and has shown antibacterial activity against different bacterial species including multi-resistant bacteria. In this study, the functional mechanism of pBD2 antibacterial activity against *Staphylococcus aureus* was investigated. After *S. aureus* cells were incubated with different concentrations of pBD2, the morphological changes in *S. aureus* and locations of pBD2 were detected by electron microscopy. The differentially expressed genes (DEGs) were also analyzed. The results showed that the bacterial membranes were broken, bulging, and perforated after treatment with pBD2; pBD2 was mainly located on the membranes, and some entered the cytoplasm. Furthermore, 31 DEGs were detected and confirmed by quantitative real-time PCR (qRT-PCR). The known functional DEGs were associated with transmembrane transport, transport of inheritable information, and other metabolic processes. Our data suggest that pBD2 might have multiple modes of action, and the main mechanism by which pBD2 kills *S. aureus* is the destruction of the membrane and interaction with DNA. The results imply that pBD2 is an effective bactericide for *S. aureus*, and deserves further study as a new therapeutic substance against *S. aureus*.

## 1. Introduction

Defensins are a family of antimicrobial peptides (AMPs) secreted by organisms. They have high biological activity against bacteria, fungi, and viruses, which makes them alternatives as novel therapeutic drugs [1,2,3]. Mammalian defensins are classified into alpha, beta, and theta defensins based on their different connection between cysteine residues, and on an evolutionary basis, beta defensins are the oldest ones [1]. In pigs, only beta defensins have been found, and all beta defensins have been discovered through genomic analysis, except for porcine beta defensin 1 (pBD1). Relatively few studies have been carried out on these defensins [4,5]. pBD1 is the first beta defensin discovered in pigs, and it has lower bactericidal activity compared with pBD2, which has strong antibacterial activity against both Gram-negative and Gram-positive bacteria including multi-resistant bacteria [6,7,8]. To date, pBD2 has perhaps the strongest antimicrobial activity known among pBDs [6,9,10]. Oral administration of pBD2 reduces the inflammatory response in weaned piglets infected with *E. coli* [11], and pBD2 attenuates inflammation and mucosal lesions in dextran sodium sulfate-induced colitis in mice [12]. Transgenic pigs and mice that constitutively overexpress pBD2 show enhanced resistance to bacterial infection [13,14]. Moreover, it has high salt-resistance and low hemolytic activity [7]. These results make it a good candidate as an antibiotic.

However, for the development of pBD2 as an antibiotic agent, it is important to understand its antimicrobial mechanism against pathogens. Many studies have proposed several patterns of antimicrobial mechanisms and focused on the cell membrane [15,16,17]. It is believed that the first step in killing bacteria is the interaction between the positively charged residues of defensin and the negatively charged components in the microbial membrane, resulting in the disruption of the cell membrane. After damaging the extracellular membrane, defensins further bind to the protoplast membrane, and cell death results directly from membrane destruction and decomposition, or changing permeability of the cytoplasmic membrane, or attacking internal targets such as negatively charged DNA or RNA, etc. [18,19,20,21,22]. Antibacterial mechanisms may vary depending on the different AMP or bacterial species [17,23,24]. There is poor experimental evidence to indicate which model is applicable to beta defensins and the mechanisms of pBDs are unclear and seldom reported.

*Staphylococcus aureus*, as an example of Gram-positive bacteria, is considered a global threat to human and animal health [25]. In a previous report, our research team cloned the pBD2 gene and constructed the engineering strain *E. coli* BL21(DE3)-pET-*pBD2* and reported that the purified recombinant pBD2 with His-Tag had strong antimicrobial activity against *S. aureus* [7]. In this study, we observed morphological changes in *S. aureus* and detected locations of action of pBD2 after incubating with recombinant pBD2. Moreover, differentially expressed genes (DEGs) were obtained, and their functions were analyzed. Our data suggest that pBD2 might have multiple modes of action, and the main mechanism by which pBD2 kills *S. aureus* is the destruction of its membrane and interaction with DNA. The results may contribute to future efforts aiming to develop this AMP as a new therapeutic substance against *S. aureus* infections.

## 2. Results

### 2.1. Antibacterial Activity of pBD2

The fused pBD2 was induced and purified as described previously using the constructed engineering strain *E. coli* BL21(DE3)-pET-*pBD2* in our lab [7]. The purified recombinant pBD2 with a His-Tag had high purity (shown in Figure 1a). The survival percentage of *S. aureus* (10^9^ cfu/mL) decreased with increasing pBD2 concentrations (*p* < 0.01), while there was no obvious decrease with exposure time (*p* > 0.05), which implies that pBD2 has high antimicrobial activity against *S. aureus* (Figure 1b). 

### 2.2. Morphological Changes in Bacteria

Morphological changes in *S. aureus* were observed by scanning electron microscopy (SEM) and are shown in Figure 2. Morphological changes in response to different pBD2 concentrations were very similar, except for those in cells incubated with 37.5 μg/mL pBD2 for 4 h. The control cells were normal without obvious damage (Figure 2a,e). After *S. aureus* was incubated with pBD2 for 1 h, cells were bulging, and the debris of bacteria was highly visible (Figure 2b–d). After incubation for 4 h, cells showed more bulging (Figure 2h), and more debris of bacteria was visible (Figure 2g,i). The cells were rough, extremely pitted, and perforated after incubation with 37.5 μg/mL pBD2 for 4 h, and these changes were different from those observed in other pBD2-treated cells (Figure 2f). However, some cells did not have obvious changes (Figure 2b,c,g). Furthermore, the bacterial biomass decreased with increasing pBD2 concentrations when bacteria were collected after they were treated with pBD2, and the number of cells in the total observed field of view decreased with increasing doses of pBD2, which indicates that more cells were dead with higher concentrations.

### 2.3. Localization of pBD2 Peptides

The location of pBD2 was determined by immuno-fluorescence microscopy (IFM) and is shown in Figure 3. The green fluorescence from staining with FITC (fluorescein isothiocyanate) labeled secondary antibodies was mainly concentrated on the surface of *S. aureus*, and distinct rings were formed, which indicates that pBD2 is mainly located on the bacterial membrane, and some entered the cytoplasm (Figure 3e,f). In addition, about 65.7% of cells were labeled with green fluorescent.

Moreover, precise localizations of pBD2 were further determined by immuno-gold transmission electron microscopy (TEM) after cells were cultured with 150 μg/mL pBD2 for 4 h (Figure 4). pBD2 marked with gold particles was mainly located on the cell membranes, and some had entered the cytoplasm, in other cells, more pBD2 was inside of the cytoplasm than on the cell membrane, without obvious membrane disruption (Figure 4b). The results from both approaches indicate that pBD2 mainly localized to the membrane and that some entered the cytoplasm.

### 2.4. Identification of Differentially Expressed Gene Fragments by ACP (Annealing Control Primer)-Based RT-PCR 

RNAs of *S. aureus* were extracted after cells were cultured with different pBD2 concentrations. The OD_260_/OD_280_ ratios were between 1.8 and 2.2, and RNA concentrations were above 1000 μg/mL, according to spectrophotometer analysis. The differentially expressed gene fragments were obtained by ACP-based RT-PCR, and were generally the same after different treatments for each arbitrary primer. A subset of the results is shown in Appendix A. The fragments were cloned and sequenced.

### 2.5. Analysis of DEGs

The sequences were analyzed using the ENSEMBL and NCBI databases, and 31 different differentially expressed gene fragments were obtained. Besides two intergenic fragments and two unknown genes (hypothetical proteins), the other 27 fragments corresponded to 31 possible functional genes and two intergenic sequences (listed in Table 1). As there are no available genome databases of *S. aureus* ATCC 25923, five sequences were found to correspond to different genes or intergenic sequences in different *S. aureus* strains (underlined in the sequence column in Table 1). The 31 possible known DEGs revealed that 10 genes were related to ion or protein transmembrane transport in the membrane, and all 10 were integral components of the membrane; 11 genes were related to the transfer of genetic information including DNA repairs, DNA replication, DNA transcription, transcriptional regulation, and translation in the cytoplasm; and 10 genes were related to glucose, lipid, and protein metabolism in the membrane or cytoplasm (listed in Table 2). In total, 13 genes were in the membranes and 12 genes were in the cytoplasm; the locations of the other five genes are uncertain, but their molecular functions suggest that they are in the cytoplasm. There were four genes next to two differentially expressed intergenic fragments and were involved in biofilm and spore formation. Taken together, pBD2 has multiple actions that affect cell transmembrane transport, the transport of inheritable information, and some metabolic processes, but its main effects are on transmembrane transport.

### 2.6. qRT-PCR Confirmation for Selected Genes 

qRT-PCR was used to confirm the DEGs. The relative expression of each gene was normalized to glyceraldehyde-3-phosphate dehydrogenase (GAPDH) gene expression. The results of qRT-PCR are consistent with those of the ACP-based RT-PCR (Figure 5 and Table 1).

The expression of genes for transport is shown in Figure 5a. After cells were treated with pBD2 for 1 h, all of these genes had lower expression levels than those of the control and showed significant differences, except for *mnhD* (*p* < 0.01). After cells were treated for 4 h, *MFS* was upregulated, and the expression levels of other genes were lower, except for those of *isdF, qoxA* and *MP*, which were upregulated in response to 37.5 μg/mL pBD2.

The expression of genes related to metabolism is shown in Figure 5b. *Laccase* was generally significantly upregulated. pBD2 treatment decreased the expression of *fadD, glmM, lacG, moaA*, and *thiE*, except for the treatment at 37.5 μg/mL pBD2 for 4 h (*p* < 0.01 or *p* < 0.05). The expression levels of *clpB* and *hmgA* were higher with treatment for 4 h and at 37.5 μg/mL pBD2 for 1 h.

The genes related to the transport of inheritable information were generally downregulated and are shown in Figure 5c. The genes related to DNA repair were generally downregulated, except for at the lower concentration for 4 h. The expression levels of genes related to DNA transcription/translation were 0.01–0.6-fold lower than those of the control, except for those of *topB* and *infB*, which had higher expression at 37.5 μg/mL pBD2 for 4 h (*p* < 0.01).

The genes next to two differentially expressed intergenic fragments were selected and detected (Figure 5d). *glmU, spoG, ipk*, and *veg* were downregulated, except for *glmU* and *ipk* at 37.5 μg/mL pBD2 for 4 h (*p* <0.01). The expression of *hp3*, located beside the possible intergenic sequence, also changed significantly (*p* < 0.01). Whether the sequences regulated the expression of nearby genes was not confirmed, although their nearby genes were changed. *hp2* expression was obviously changed, and its function needs further study.

### 2.7. Gel Retardation Assay of pBD2 Binding to DNA 

The ability of pBD2 to bind DNA was assessed by a gel retardation assay. pBD2 at different concentrations was fused with DNA for 1.5 h at room temperature, and then DNA was separated by agarose gel (Appendix A). pBD2 bound to DNA and induced gel retardation at all concentrations tested. The DNA was significantly reduced with increasing pBD2 concentrations; the pBD2-bound DNA was trapped, and its migration through pores was retarded. The results suggest that pBD2 has DNA-binding activity.

## 3. Discussion

pBD2 has high antibacterial activity, which makes it an ideal candidate as an antibiotic [7,8]. Our previous reports showed that pBD2 with His-Tag had high antibacterial activity against *S. aureus* including multi-resistant bacteria [7]. The bacterial concentration for antimicrobial tests was usually approximately 10^4^–10^6^ cfu/mL. In this study, 10^9^ cfu/mL *S. aureus* was used because a large number of bacterial cells were needed in subsequent experiments. It was surprising that purified recombinant pBD2 had the same strong bactericidal ability, and antibacterial activity increased with pBD2 concentrations (*p* < 0.01), which is consistent with the results of experiments using the 10^5^–10^6^ cfu/mL bacteria reported previously by our lab [7].

The morphological changes in *S. aureus* were very similar after incubation with different pBD2 concentrations (Figure 2). The morphological changes indicated some extent of pBD2-induced damage to cell membranes. However, some cells did not have obvious damage, which may indicate that pBD2 has a clustering effect and kills *S. aureus* cells one by one, and about 65.7% of cells were labeled with green fluorescent by IFM, confirming the results. Our results are consistent with the effects of the antimicrobial peptides gramicidin S and PGLa [26]. In addition, the bacterial biomass collected after treatment with higher pBD2 concentrations was lower compared with those at lower concentrations, and the debris in the total observed field of view was more visible at higher pBD2 concentrations, which indicates that cells were dead.

According to IFM, and further confirmed by immune-gold TEM, pBD2 mainly localized to the membrane and also entered the cytoplasm of *S. aureus*. In addition, cells in which more pBD2 entered the cytoplasm than localized to the cell membrane did not show obvious membrane disruption (Figure 4b). The main reason may be that lipid–peptide interaction in live bacteria is a dynamic process that causes transient disruption of the cell membrane without destroying its structural integrity [27]. Some studies have reported that AMPs mainly acted on the membrane and led to cell death such as T9W, tachyplesin I, cathelicidin, and bovine lactoferricin derivatives [28,29,30,31,32]. Some results indicated that AMPs had different locations for killing different species of microbes such as Bac7 and SpHyastatin. Bac7 mainly localized to the cell surface of *P. aeruginosa* and killed bacteria, while it targeted protein synthesis when acting on other Gram-negative bacteria [24,33]. SpHyastatin acted specifically on the surface of *S. aureus*, whereas it could enter the cytoplasm of *P. fluorescens* [34]. Furthermore, some results showed that antimicrobial peptides had different locations, even if they had similar structures such as buforin II, Indolicidin, and their analogs [20,35]. Buforin II with a P11A mutation localized to the membrane, while buforin II entered the cells [35]. Indolicidin (IN), and its analogs IN-1 and IN-2 were detected on the membranes of *E. coli*, whereas IN-3 and IN-4 penetrated the membranes [20]. In this study, pBD2 was mainly located on the membrane, and some entered the cytoplasm, which may imply that pBD2 acts on the membrane and internal targets.

In this study, RNA was not extracted by phenol, which causes specific losses of polyadenylated-RNA [36]. Therefore, the results prove that the prokaryotic RNA obtained by ACP-based RT-PCR contained polyadenylic acid without adding poly(A). ACP-based RT-PCR technology is an easy differential genes analysis technology that uses agarose gel electrophoresis, which is widely used to identify DEGs in response to different conditions [37,38,39]. In this study, ACP-based RT-PCR was used to detect DEGs in response to pBD2 treatments. With this technique, DEGs were detected and found to be highly similar in response to different pBD2 concentration treatments for each annealing control primer, which means that ACP-based RT-PCR used in this study is credible and repeatable. Furthermore, the results of ACP-based RT-PCR were confirmed by those of qRT-PCR. In this study, 31 DEGs were obtained: the 31 possible known functional genes are involved in transport, DNA, and some metabolic processes, but they are mainly related to transport through the membrane.

Our results indicate that transmembrane transport was strongly influenced and accounted for one- third of DEGs. Among the 10 transporters, bcrB, isdF, and lysP belong to the ATP-binding cassette (ABC) superfamily protein; qoxA and MFS belong to the major facilitator superfamily. These two ubiquitous families of transporters account for nearly half of the solute transporters encoded within the genomes of microorganisms, and they transport a wide variety of substrates including ions, amino acids, peptides, proteins, lipids, sugars, and so on across extra- and intracellular membranes to import nutrients or to export waste and toxic products [40,41]. Aside from these, qoxA, mnhA, and mnhD are involved in ATP synthesis-coupled electron transfer. The expression of these genes significantly changed, indicating that transmembrane transport and energy states were strongly influenced. Using microarray analysis, Sass et al. reported that the modes of action of hBD3 against *S. aureus* were related to ABC transporters, and knockout of the transporter genes significantly enhanced susceptibility to hBD3 [42]. Antimicrobial peptide LL-37 has been reported to kill bacteria mainly by targeting energy metabolism [19]. Wenzelet al. reported that antimicrobial peptides had impacts on the membrane including the inhibition of the respiratory chain and a reduction in ATP levels [43]. kdpB is an important part of potassium ion (K+) transport, which is a critical determinant of growth and survival through the regulation of cytoplasmic pH and cell structure; a loss of cytoplasmic potassium leads to cell death [44,45]. It was reported that defensins could inhibit potassium channels, which was one of the antimicrobial mechanisms [46,47]. The changes in transmembrane transporters influenced transport, energy states, ion balance, and so on, which are all causes of cell death.

Eleven genes were related to the transport of genetic information. It has been reported that antimicrobial peptides interact with intracellular molecules such as DNA, which can subsequently influence DNA or RNA synthesis or protein synthesis to kill bacteria, especially proline-rich antimicrobial peptides [48,49]. N4, Oncocin and its derivatives and so on were reported to bind specifically to DNA, disrupt DNA conformation and inhibit DNA, RNA or protein synthesis [22,50,51]. Tryptophan-rich antimicrobial peptides also have this mechanism [52]. Our gel retardation assay results show that pBD2 could bind to DNA. pBD2 interacted with DNA, and according to the functional analysis of DEGs, it affected all the transfer processes of genetic information. The expression levels of these genes were generally downregulated (*p* < 0.01), except for those of *uvrB*, *MutL*, *topB*, and *infB*, which were upregulated after treatment with 37.5 μg/mL pBD2 for 4 h, especially *uvrB* and *MutL*, which may mean that these surviving cells repaired their damage in this condition, but this also needs further research.

The genes for metabolic processes were also influenced, and generally downregulated (*p* < 0.01). These were related to carbohydrate, lipid, and protein metabolism and mainly related to biosynthetic processes, especially membrane biogenesis. fadD, glmM, and oal are involved in the biogenesis of the membrane. Moreover, glmU and spoG, ipk and veg (next to two differentially expressed intergenic fragments) mainly influence membrane biogenesis and cell proliferation. It was reported that human beta defensin 3 (hBD3) killed *S. aureus* by inhibiting cell wall biosynthesis [53]. Dosunmuet al. reported that peptide TP359 decreased the expression of outer membrane biogenesis genes in *Pseudomonas aeruginosa*, which reduced the organism’s structural stability and led to cell death [54]. Synthetic cationic peptide 1037 also significantly reduced the biofilm formation of *Pseudomonas aeruginosa* and led to cell death [55]. Antimicrobial peptidomimetics (SAMPs) and Hyicin 4244 acted on staphylococcal biofilms [56,57]. Antimicrobial peptides C18G (Anaspec) blocked cell division [58].

Five fragments corresponded to more than one gene, and the expression levels of these genes changed markedly according to qRT-PCR, which was selected for confirmation. *InfC, rep*, and *isdF* corresponded to the same DEG, and they all changed significantly. However, based on the consistency of qRT-PCR and ACP-based RT-PCR results, the DEG is likely to be *isdF*, and not *infC* or *rep*. *araC* and *lysP* also corresponded to the same DEG; the results of qRT-PCR for both genes were consistent with those of ACP-based RT-PCR, and it was difficult to distinguish between them. The other three sequences were also difficult to distinguish and need further analysis. In addition, genes next to differentially expressed intergenic fragments were detected; although it is not directly evident that the sequences regulated the expression of their nearby genes, the expression of their nearby genes changed.

From the analysis above, pBD2 has multiple and variable modes of action. It acted on the cell membranes, affected DNA and influenced biosynthetic processes, among other actions. Overall, the main action of pBD2 is the disruption of the cell membrane and interaction with DNA.

## 4. Materials and Methods

### 4.1. Preparation of Porcine Beta Defensin 2

Based on the constructed engineering strain *E. coli* BL(DE3)-pET-*pBD2* constructed in our laboratory, recombinant pBD2 was induced and purified by the His-Tag affinity column according to the procedure reported previously [7]. The concentrations were determined by the bicinchoninic-acid (BCA) method.

### 4.2. Bacteria with pBD2 Treatment

The killing kinetics of pBD2 were determined by the turbidimetric method [59]. Overnight cultured *S. aureus* ATCC 25923 was transferred into fresh medium, cultured at 37 °C, and 220 rpm for 2–4 h until the OD_600_ value was about 1 (bacteria were in the logarithmic phase, and the density was about 10^9^ cfu/mL), and then *S. aureus* ATCC 25923 cells were incubated with different pBD2 concentrations in a polypropylene 96-well microtiter plate for different times. The growth of *S. aureus* was measured by optical density (OD) at dual wavelengths (630 nm and 405 nm) by a microplate reader (Stat Fax 2100, Awarenss Technology Inc., Palm, FL, USA). The survival percentages were calculated and defined as the ratio of cell density in the presence of pBD2 to the cell density without pBD2, according to a previous report [59]. In addition, after *S. aureus* cells were incubated with pBD2 at 37 °C for 1 h or 4 h in tubes, the cells were harvested, washed, and then prepared for scanning electron microscopy (SEM), immuno-fluorescence microscopy (IFM), immuno-gold transmission electron microscopy (TEM), or RNA extraction.

### 4.3. Morphological Changes in S. aureus

Morphological changes in *S. aureus* were observed according to previous reports with some modifications [26,28]. In brief, after pBD2 treatment, *S. aureus* cells were fixed with 2.5% glutaraldehyde overnight at 4 °C, subsequently dehydrated with gradient concentrations of ethanol, and then placed in an ethanol and tert-butyl alcohol mixed solution (1:1). The treated samples were dropped on tin foil, and dried in the air. Sample were gold-coated by an ion spray instrument (MSP-2S, IXRF, Austin, TX, America) and then observed by environmental scanning electron microscopy (FEI Quanta 250, Hillsboro, OR, USA).

### 4.4. Localization of pBD2 Peptides

The locations of pBD2 were detected by IFM according to a previous report with some modification [60]. In brief, *S. aureus* cells were fixed with 4% paraformaldehyde for 2–4 h at 4 °C on anti-off slides (Wuhan Boster Biological Engineering Co., Ltd, Wuhan, China), and treated with 0.5% Triton-100 for 15 min. After incubation with 10% normal goat serum and then further incubated with the pBD2 polyclonal antibody (prepared by our group from rabbit [61], diluted to 1:500) at 4 °C overnight, these cells were finally incubated with FITC (fluorescein isothiocyanate)-labeled secondary antibodies for 1 h at 37 °C in the box. After that, cells were dyed by DAPI (4′,6-diamidino-2-phenylindole) solution for 5 min in the dark, blocked by a blocking agent containing anti-fluorescent quencher, and sealed with cover slips. Controls for IFM were performed by incubation with phosphate buffer (PB) instead of primary antibodies. The cells were observed by immunofluorescence microscopy (DM6000B, Leica, Wetzlar, Germany).

Immuno-gold TEM was used to further determine the localization of pBD2 and performed by the method reported previously [60]. The cells were embedded in 2% low-melting- point agarose and fixed with 3% paraformaldehyde (pH 7.4), dehydrated with gradient concentrations of methanol and finally infiltrated in pure Lowicryl K4M. The samples were cut into 0.1 μm thick sheets by a frozen slicer (CM1900, Leica, Wetzlar, Germany). Sections were incubated with 0.05% Triton X-100, 0.05% Tween, and 1% bovine serum albumin (BSA) in 0.01 M phosphate-buffered saline (PBS, pH 7.4) for 5 min at 20 °C. Then, they were incubated with the pBD2 antibody (1:500) for 2 h and protein-A gold (1:50) for 1 h. Finally, specimens were stained with 3% uranium acetate for 4 min and lead citrate for 1 min and observed by electron microscope (Tecnai F12 S-Twin, FEI, USA). The results from the control cells were obtained by incubating cells with PBS buffer without primary antibodies.

### 4.5. RNA Extraction

The total RNAs of *S. aureus* were extracted according to the procedure reported previously [36]. The extracted total RNA was treated with DNase for 30 min and purified again. The quality of RNAs was detected by 1% agarose gel electrophoresis, and the concentrations were determined by UV absorption using a spectrophotometer (Nanodrop 2000/2000C, Thermo Scientific, Waltham, MA, USA).

### 4.6. Identification and Analysis of DEGs

DEGs were identified according to the manufacturer’s instructions for the GeneFishing^TM^ DEG Kit using 20 different arbitrary primers in combination with an oligo-dT anchor primer (Seegene, Seoul, South Korea). The differentially expressed gene fragments were obtained by ACP-based RT-PCR; the amplified PCR products were separated on 1% agarose gel, and DEGs were extracted from the gel using a DNA Purification Kit (TIANGEN, BeiJing, China) and directly cloned and sequenced. Sequences were compared with the databases in GenBank and Ensembl using the BLAST programs to find their functions.

### 4.7. Quantitative Real-Time PCR

qRT-PCR was performed with SYBR green dye, according to the manufacturer’s protocol (TaKaRa, Dalian, China) using a qRT-PCR platform (LightCycler 96, Roche, Base, Switzerland). All samples were analyzed in triplicate, and values were calculated by the 2^−^^△△CT^ methods with GAPDH as a reference gene.

### 4.8. Gel Retardation Assay

A gel retardation assay was performed as described previously with some modifications [62]. In brief, 10 μL of different concentrations of pBD2 (0, 37.5, 75 and 150 μg/mL) were incubated with 10 μL of 129.2 ng/μL DNA for 1.5 h at room temperature, Then, 5 μL of the reaction solution was mixed with 2.5 μL of 6X gel loading buffer, and electrophoresed on 1% agarose gel for 30 min at 100 V.

### 4.9. Statistical Analysis 

Analysis of variance (ANOVA) was performed with the software SPSS 17.0 (IBM, Armonk, NY, USA). LSD or Dunnett’s method was used to compare treatment means. Significance and extremely significance were defined as *p* < 0.05 and *p* < 0.01, respectively.

## 5. Conclusions

After *S. aureus* was cultured with different pBD2 concentrations, the morphology of *S. aureus* changed markedly, and the debris was highly visible. According to IFM and immune-gold TEM, pBD2 mainly localized to the cell membrane, and some entered the cytoplasm. The detected DEGs were related to transmembrane transport, DNA, and some metabolic processes. In addition, they were generally predominantly downregulated. In sum, the main mechanism by which pBD2 kills *S. aureus* is the destruction of cell membranes and interaction with DNA. Our study contributes to understanding the mechanism of pBD2 and lays the foundation for further research on its application in practice.

## Figures and Tables

**Figure 1 molecules-25-03513-f001:**
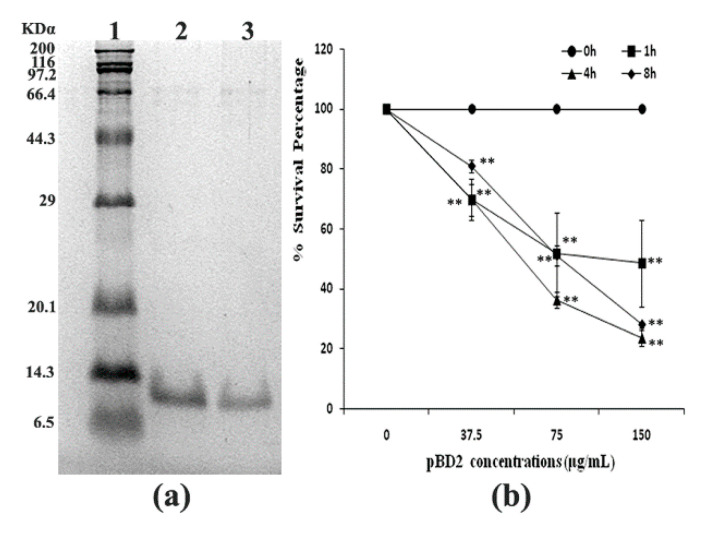
Purified fused pBD2 and its antimicrobial activity. (**a**) SDS-PAGE analysis of purified fused pBD2. Lane 1 indicates the protein marker; Lanes 2–3 indicate purified pBD2. (**b**) Antimicrobial activities of recombinant pBD2 against *S. aureus* at different time points. The ** symbol indicates extremely significant differences compared with the control at *p* < 0.01 by Dunnett’s method.

**Figure 2 molecules-25-03513-f002:**
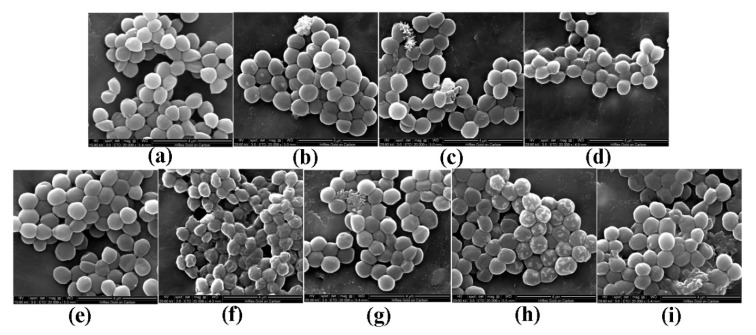
Scanning electron microscopy (SEM) images of morphological changes in bacteria after they were cultured with pBD2. (**a**) Control cells incubated without pBD2 for 1 h; (**b**–**d**) cells were treated for 1 h with 37.5, 75, and 150 μg/mL pBD2 respectively; (**e**) control cells incubated without pBD2 for 4 h; (**f**,**g**) cells were treated for 4 h with 37.5 and 75 μg/mL pBD2, respectively; (**h**,**i**) cells were treated for 4 h with 150 μg/mL pBD2.

**Figure 3 molecules-25-03513-f003:**
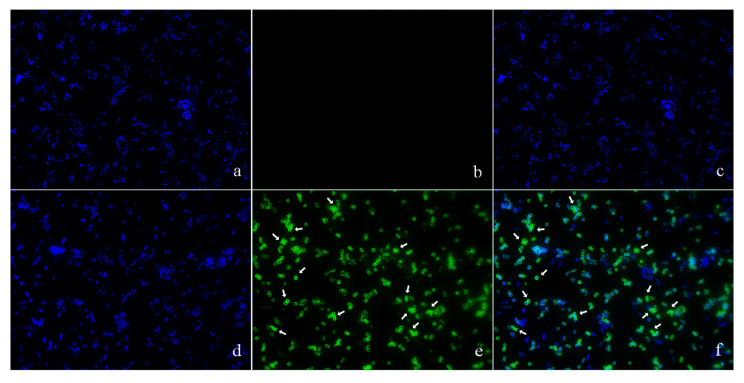
The locations of pBD2 on bacteria were detected by IFM (×1000). (**a**–**c**) Control bacteria (without the first antibody) dyed by DAPI (4′,6-diamidino-2-phenylindole), FITC, and DAPI/FITC, respectively; (**d**–**f**) bacteria dyed by DAPI, FITC, and DAPI/FITC, respectively. Bacteria were magnified 1000 times. The green fluorescence, which is the result of staining with FITC-labeled secondary antibodies, indicates the location of pBD2. The blue fluorescence, which is the result of staining with DAPI, indicates the location of bacterial DNA. The arrows indicate the cells in which pDB2 is likely to be located inside.

**Figure 4 molecules-25-03513-f004:**
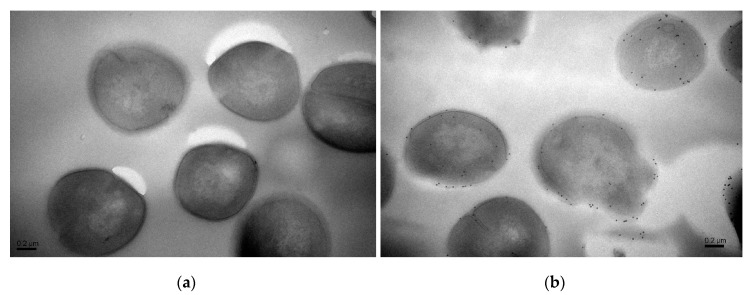
The locations of pBD2 on bacteria were detected by immuno-gold TEM. (**a**) Control cells; (**b**) cells after they were treated with 150 μg/mL pBD2 for 4 h. The gold particles labeling the secondary antibody indicate the location of pBD2.

**Figure 5 molecules-25-03513-f005:**
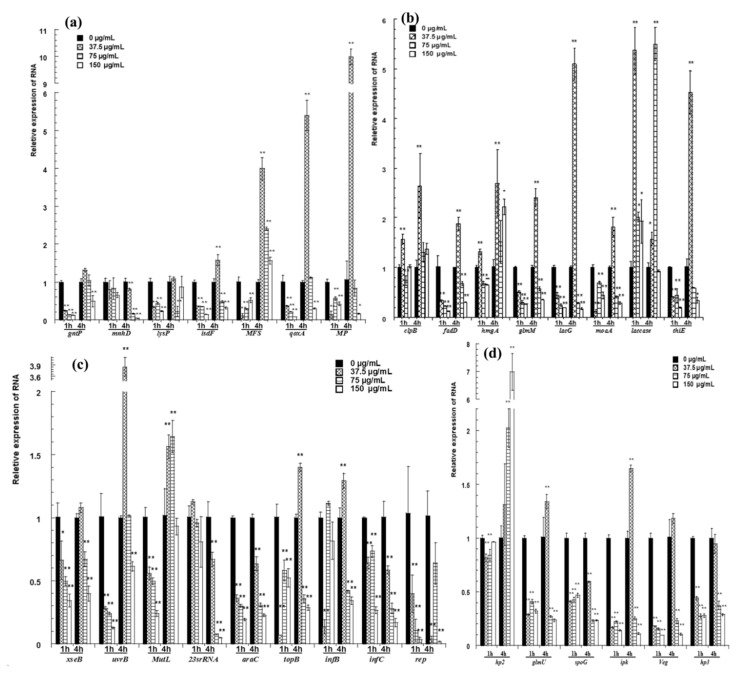
qRT-PCR confirmation for selected genes. (**a**) The relative expression of genes for transport; (**b**) the expression of genes related to metabolism; (**c**) the relative expression of genes related to transport of inheritable information; (**d**) the relative expression of hypothetical proteins and the genes next to two differentially expressed intergenic fragments. The changes in gene expression were calculated by the 2^−^^△△CT^ method with GAPDH as a reference gene. The * and ** symbols at the top of the column indicate significant differences from the control at *p* < 0.05 and *p* < 0.01, respectively.

**Table 1 molecules-25-03513-t001:** The differentially expressed gene fragments determined by the GeneFishing DEGs kit.

Sequence ^1^	GeneFishing	Size (bp)	Protein
(371h 3-2)/(/37S4h 3-1/3-2/16/75S4h 7-1)	up/down	193/196/300/209/188	23S ribosomal RNA.
37S1h 15-2	down	84	DNA mismatch repair protein MutL.
(37S1h/37S4h 20-1/75S1h/150S4h 20-1)	down/up	228/219/220/216	Formate-tetrahydrofolate ligase.
37S4h 5	up	52	Gluconate transporter/gluconate permease.
37S4h 7-2	up	73	6-Phospho-beta-galactosidase; or intergenic sequence: one is a transposase family protein; the other one is a hypothetical protein (378 bp, 215 aa, HP3).
37S4h 9-1	up	273	Protein disaggregation chaperone/ATP-dependent chaperone protein ClpB.
37S4h 11-2	up	24	Membrane protein.
37S 4h 12-2/12-1	up	34/29	MFS transporter.
37S4h 15-1	up	39	Molybdenum cofactor biosynthesis protein A.
37S4h 15	up	88	Heme ABC transporter permease; or translation initiation factor IF-3; or replication-associated protein.
75S1h 1	down	57	RNA polymerase sigma factor RpoD.
75S1h 4-1	down	26	AraC family transcriptional regulator; or gamma-aminobutyrate permease.
75S1h 5-1	down	199	3-Hydroxy-3-methylglutaryl-CoA reductase.
75S1h 9	up	82	Laccase.
75S4h 11-1	down	124	DNA topoisomerase III.
75S1h 14	down	213	Potassium-transporting ATPase C chain.
75S4h 15-1	down	87	Phosphoglucosamine mutase.
75S4h 18-1	down	232/228	Long-chain fatty acid--CoA synthetase.
150S1h 15-2	down	70	Exodeoxyribonuclease VII, small subunit.
150S4h 1	down	60	Na(+) H(+) antiporter subunit A/monovalent cation/H+ antiporter subunit A.
150S4h 4-1	down	130	Quinol oxidase polypeptide II, integral component of membrane.
150S4h 5	down	166	Translation initiation factor 2.
150S4h 6	up	115	O-Antigen ligase family protein or intergenic sequences: both are hypothetical proteins.
150S4h 5/19	down	80/105	Thiamine-phosphate pyrophosphorylase.
150S4h 9	down	84	Zn-dependent hydrolase.
150S4h 9-1	down	98	Na(+) H(+) antiporter subunit D/monovalent cation/H antiporter, subunit D.
150S4h 9-2	down	54	Excinuclease ABC subunit B or bacitracin ABC transporter permease.
37S1h 15-1	up	60	Intergenic sequence: one is glucosamine-1-phosphate acetyltransferase, the other is stage V sporulation protein G.
150S4h 9-3	up	50	Intergenic sequence: one is veg protein; the other is 4-diphosphocytidyl-2C-methyl-D-erythritol kinase.
37S4h 20-2	up	136	Hypothetical protein 1 (366 bp, 121 aa).
75S1h 5-2	down	80	Hypothetical protein 2 (396 bp, 131 aa).

^1^ The first number of the sequence represents the cultured pBD2 concentrations (i.e., 37.5, 75, or 150 μg/mL); S is the abbreviation of *S. aureus*; 1 h or 4 h indicates the exposure time with pBD2. The last two numbers indicate the number of random ACPs in the GeneFishing DEG kit and amplified bands. The underlined sequences indicate correspondence to two or more possible genes.

**Table 2 molecules-25-03513-t002:** The positions, functions, and abbreviations of functional genes.

Name	Main Functions	Positions	Abbreviations
**Transporter**			
Na(+) H(+) antiporter subunit A	involved in ATP synthesis-coupled electron transfer	integral component of membrane	mnhA
Na(+) H(+) antiporter subunit D	involved in ATP synthesis-coupled electron transfer	integral component of membrane	mnhD
quinol oxidase polypeptide II	ATP synthesis-coupled electron transport chain, transport; respiratory chain oxidoreductase activity	integral component of membrane/plasma membrane	qoxA
gamma-aminobutyrate permease	amino acid transmembrane transport	integral component of membrane	lysP
gluconate transporter	gluconate transmembrane transport	integral component of membrane	gntP
MFS transporter	transmembrane transport	integral component of membrane	MFS
bacitracin ABC transporter permease	transport	integral component of membrane	bcrB
potassium-transporting ATPase C chain	potassium transport	integral component of membrane/plasma membrane	kdpB
membrane protein	transport	integral component of membrane	MP
heme ABC transporter permease	transmembrane transport	integral component of membrane/plasma membrane	isdF
**DNA Repair, Transcription, and Translation**			
exodeoxyribonuclease VII, small subunit	exonucleolytic cleavage, DNA repair	in cytoplasm	xseB
excinuclease ABC subunit B	DNA repair, nucleotide excision repair, SOS response	in cytoplasm	uvrB
DNA mismatch repair protein MutL	components of mismatch repair complex, repair of mismatches in DNA	in cytoplasm	MutL
RNA polymerase sigma factor RpoD	transcription initiation from bacterial-type RNA polymerase promoter	in cytoplasm	rpoD
23S ribosomal RNA	protein synthesis	in cytoplasm	23srRNA
DNA topoisomerase III	releases the supercoiling and torsional tension of DNA during the DNA replication and transcription	in chromosomes	topB
AraC family transcriptional regulator	transcriptional regulator	in cytoplasm	araC
translation initiation factor 2	the initiation of protein synthesis	in cytoplasm	infB
translation initiation factor 3	the initiation of protein synthesis	in cytoplasm	infC
replication-associated protein	replication-associated	in cytoplasm	rep
Zn-dependent hydrolase	RNA processing; RNA phosphodiester bond hydrolysis	unknown, possibly in cytoplasm	zdh
**Metabolism**			
protein disaggregation chaperone/ATP-dependent chaperone protein ClpB	nucleoside triphosphatase activity disaggregates misfolded and aggregated proteins; cell recovery from heat-induced damage.	in cytoplasm	clpB
long-chain fatty acid--CoA synthetase	involved in fatty acid and lipid metabolism, phospholipid biosynthetic process	inner membrane	fadD
3-hydroxy- 3-methylglutaryl-CoA reductase	rate-controlling enzyme of the mevalonate pathway, non-sterol isoprenoids biosynthetic process	integral component of membrane	hmgA
phosphoglucosamine mutase	carbohydrate metabolic process, participates in both the breakdown and synthesis of glucose	unknown, possibly in cytoplasm	glmM
6-phospho-beta-galactosidase	carbohydrate metabolism, lactose degradation	unknown, possibly in cytoplasm	lacG
molybdenum cofactor biosynthesis protein A	involved in the pathway of molybdopterin biosynthesis, redox action	molybdopterin synthase complex, possibly in cytoplasm	moaA
Laccase	formation or degradation of lignin	unknown, possibly in cytoplasm	laccase
thiamine-phosphate pyrophosphorylase	participates in thiamine metabolism,	unknown, possibly in cytoplasm	thiE
	thiamine diphosphate biosynthesis		
O-Antigen ligase family protein	ligase activity, biogenesis of the outer membrane	integral component of membrane	oal
formate-tetrahydrofolate ligase	participating in the transfer of one-carbon units, an essential element of various biosynthetic pathways	in cytoplasm	fhs
**Protein beside Intergenic Sequences**			
glucosamine-1-phosphate acetyltransferase	plays an important role in maintenance of cell shape, involved in lipopolysaccharide and peptidoglycan biosynthetic processes	in cytoplasm	glmU
stage V sporulation protein G	participation in the barrier formation of spores	unknown	spoG
4-diphosphocytidyl-2C-methyl-D-erythritol kinase	terpenoid biosynthetic process	unknown	ipk
	isopentenyl diphosphate biosynthetic process		
Veg protein	biofilm formation	unknown	Veg
hypothetical protein 3 (378 bp, 215 aa)	unknown	unknown	hp3

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
