# Peer review of "Antimicrobial Mechanism of pBD2 against Staphylococcus aureus"

_molecules, 2020, doi:10.3390/molecules25153513_

Round 1

Reviewer 1 Report

Authors describe the analyses of the functional mechanism of porcine beta defensin 2 (pDB2), an antimicrobial peptide against Staphylococcus aureus. The pDB2 has exhibited an antibacterial activity against bacteria including multi-drug resistant ones, therefore, it could be a potential therapeutics against those causing infectious diseases. They carried out various experiments and obtained interesting results. However, some results and discussion seem to be ambiguous and should be clarified before publication.  

Figure 1.The concentration of SDS-PAGE gel should be adjusted so that the target band comes upper, and the molecular weight maker including smaller bands should be used so that the target band locates between marker bands.

Figure 3. Authors should make enlarged insets of figures. Since each cells are too small to see if they are labeled as distinct rings. They also should indicate some example cells in which pDB2 located inside by using such as arrow heads.

Figure 4. Authors described that gold particles were mainly on the cell membranes, and some in cytoplasm. However, in the normal shape cells (upper right and center left) have more particles inside of cells than in the irregular shape cell (center). The explanation should be added regarding this.

Figure 7 and Supplementary Fig. 7.

Authors carried out the gel retardation assay. In the legend, the concentrations of pDB2 seems to be decreasing from lane 1 to 4. Is this what they meant? DNA bands appeared in the gel are pDB2-bound or free? In a retardation assay, it is expected two bands, bound and free, in each lane. Otherwise, it is impossible to conclude whether the binding occurred even if the band intensities in the gel changed.

Regarding DEGs experiments, 

Line 239, This reviewer does not understand why the lowered expression of the membrane transport genes related with physically disrupted membrane. The transport proteins are a part of membrane proteins, but not scaffolds, so how their decrease links to the disruption of the membrane. Also, if pDB2 disrupts the cell membrane, why and how the gene expression of transport proteins were decreased ?

In relation with this, in line 137, authors described that pDB2 has multiple actions. The gene expression has been influenced in several cell systems, but the mode of action of pDB2 could be a single. Thus, the sentence also could be described as that a single action of pDB2 affect various physiological processes in a cell.

Line341, RNA could not be digested with DNase, so they might want to say, RNA was treated with DNAase.

General question regarding DEGs experiments. Since cells started to get cell membranes disrupted as in Figure 1, thus, molecules inside of cells, especially RNAs, are easily degraded. Therefore, lowered expressions of DEGs genes may be the reflections of the degradation of RNAs. Please add sentence(s) on this point.

Reviewer 2 Report

This paper describes the antimicrobial activity and provides insight into the possible mechanism of action of porcine beta defensin 2 against the Gram-positive bacterium S. aureus. The data presented are of interest, however several issues of concern need to be addressed.

Specific comments:

  • Fig. 1c. show killing curve in log scale
  • Fig. 2. move to SI 
  • Fig. 3. Quantify results obtained
  • Fig. 5 (electrophoresis analysis). Move to SI
  • Table 1. Organize data in genes upregulated vs downregulated and show fold-change differences in each case.
  • Table 2. Show fold-change differences in gene expression in each case.
  • cite relevant literature (doi:10.1128/AAC.00064-12) and compare with results obtained in this work. 
  • Fig. 6. Increase resolution of figure and size so it is easier to see.
  • Fig. 7. Move to SI

Reviewer 3 Report

The manuscript examines the mechanism of action of pBD-2 against S. aureus. The manuscript is generally well written but could benefit from some editing by a native English speaker. There are also a number of concerns about the data which need to be addressed before it can be accepted for publication:

1) pBD2 has 37 amino acids. So why do the gels show a product which is 14.1 kDa in weight? Clearly there is an additional fusion protein segment being expressed here, but the reader has no idea what it is. Also, is the data in Figure 1c for the fusion construct? This could explain why the concentration required for any significant killing is 150 ug/mL, as compared to the ca. 35 ug/mL reported in Veldhuizen et al. I would strongly recommend that the authors plot their data as was done in the 2008 manuscript since: i) this is a more standard way of doing it (i.e. y axis is log of CFU's) and ii) the reference they give isn't even in the list of references.

2) The data shown in Figure 2 (labelled Figure 1) is confusing. For one, Figure 2f shows more of an effect than Figure 2g, yet the concentration used in Figure 2g is double that than in Figure 2f. Also, the authors state that "the number of cells reduced with increased with doses of pBD2, which indicated cells were dead more with higher concentrations.", yet again, I would not say that there are fewer cells in Figure 2i versus 2b. The authors need to be very cautious in their statements. It is very easy to find a region where there may be fewer cells simply due to sample plating.

3) Figure 3 needs to be better explained. What is the reader looking at? What is the blue and green? Same applies to Figure 4. What are we looking at here? I have to note that the authors published a similar study to the one presented here but applied to E.coli. The format of the two publications is almost identical and the lack of explanations is the same in both.

4) The data presented in Section 2.7 clearly shows that the authors do not understand how this experiment works. Gel retardation implies that there is a concentration dependent shift in the position of the bands. This is not what is observed in Figure 7.

5) The discussion has to be completely rewritten in light of the comments made above.

6) Section 4.8: What type of DNA was used?

Minor points:

- Gram should be capitalized in all cases when referring to Gram positive and Gram negative bacteria.

Round 2

Reviewer 1 Report

Authors has revised the manuscript adequately according to the comments by the reviewer. However, this reviewer still would like to stress on the gel retardation assay in 2 points.

They should describe what kinds of DNA they used.

A photo of the full size gel as well as the current one, would be useful to show the length of DNA, which authors had provided in the previous supplementary, and references authors referred did so.

Author Response

Dear reviewers,

Thank you very much for giving us another opportunity to revise our manuscript entitled “Antimicrobial Mechanism of pBD2 against Staphylococcus aureus” (Manuscript ID: molecules-860238). We appreciate your helpful comments and suggestions. Please find our point-to-point reply below. Revised portions could be seen in the revised manuscript with trace changes. We hope that our revisions adequately address your concerns. Thank you in advance for your consideration of this revised manuscript.

Best regards,

Chunli Li

Responses to comments:

Authors has revised the manuscript adequately according to the comments by the reviewer. However, this reviewer still would like to stress on the gel retardation assay in 2 points.

They should describe what kinds of DNA they used.

Response: Thank you for your advice. The S.aureus gemonic DNA was used, we add it in the manuscript (line 205 and line 383).

A photo of the full size gel as well as the current one, would be useful to show the length of DNA, which authors had provided in the previous supplementary, and references authors referred did so.

Response: Thank you for your advice. The figure was changed with full size gel in supplementary manuscript.

Reviewer 3 Report

The authors have addressed some of the reviewer's concerns but unfortunately, some remain:

Figure 4: Where are the gold particles in the image? The brights spots in a) or the dark small dots in b)? The authors have to be clear about what the reader is looking at.

The DNA retardation results shown in Supplemental are not indicative of binding (unlike e.g. in the paper by Park et al. on buforin 2). When a complex forms, the molecular weight of the complex increases and hence the band appears at a different position. This is not what we see in Figure S2. All we see is a decrease in intensity, which can mean many other things: aggregation, incorrect loading, etc.... The authors need to remove this entire section and reword the text accordingly.

Once these issues are addressed, the manuscript may be reconsidered.

Author Response

Dear reviewers,

Thank you very much for giving us another opportunity to revise our manuscript entitled “Antimicrobial Mechanism of pBD2 against Staphylococcus aureus” (Manuscript ID: molecules-860238). We appreciate your helpful comments and suggestions. Please find our point-to-point reply below. Revised portions could be seen in the revised manuscript with trace changes. In addition, We search for the paper published by park et al. on buforin 2. one among thirteen paper was found to meet the requirement and we read it carefully ( the title is” Mechanism of Action of the Antimicrobial Peptide Buforin II: Buforin II Kills Microorganisms by Penetrating the Cell Membrane and Inhibiting Cellular Functions”, park, et al. 1998). If I didn’t find the right paper, please let me know.We hope that our revisions adequately address your concerns. Thank you in advance for your consideration of this revised manuscript.

Best regards,

Chunli Li

Figure 4: Where are the gold particles in the image? The brights spots in a) or the dark small dots in b)? The authors have to be clear about what the reader is looking at.

Response: We apologize for it. the gold particles are the dark small dots in b, and the sentences were changed into “The gold particles (the dark small dots in b) labeling the secondary antibody indicate the location of pBD2”in the revised manuscript (117-118).

The DNA retardation results shown in Supplemental are not indicative of binding (unlike e.g. in the paper by Park et al. on buforin 2). When a complex forms, the molecular weight of the complex increases and hence the band appears at a different position. This is not what we see in Figure S2. All we see is a decrease in intensity, which can mean many other things: aggregation, incorrect loading, etc.... The authors need to remove this entire section and reword the text accordingly.

Response: We search for the paper published by park et al. on buforin 2. one among thirteen paper was found to meet the requirement and we read it carefully ( the title is” Mechanism of Action of the Antimicrobial Peptide Buforin II: Buforin II Kills Microorganisms by Penetrating the Cell Membrane and Inhibiting Cellular Functions”, park, et al. 1998). if it is this paper, our result is identical to that of this paper. In this paper, the figure 3a was the gel retardation assays of DNA, and it did show one band; the figure 3b was the gel retardation assays of RNA, which showed two bands: one perhaps was 16S RNA, another one was 23sRNA (RNA should show at least two bands). DNA does not show two bands due to antimicrobial peptides (AMP) binding, nor it located in different positions. Our gel retardation result is identical to that of this paper, and other papers also showed the same results (I listed some of the papers when I responded to comments last time). In all of these papers, the authors stated that the results proved the binding of AMP to DNA. The reason may be that the positively charged residues of the AMP interact with the negatively charged DNA, and the charge is neutralized, so AMP-Boud DNA cannot migrate from the pore. If I didn’t find the right paper, please let me know.